# CD28 engagement inhibits CD73-mediated regulatory activity of CD8+ T cells

Yo-Ping Lai[1], Lu-Cheng Kuo[1], Been-Ren Lin[2], Hung-Ju Lin[1], Chih-Yu Lin[3], Yi-Ting Chen[4], Pei-Wen Hsiao[3], Huan-Tsung Chang[5], Patrick Chow-In Ko[6], Hsiao-Chin Chen[7], Hsiang-Yu Chang[5], Jean Lu[8,9,10], Hong-Nerng Ho [11,12], Betty A. Wu-Hsieh [12], John T. Kung [4] & Shu-Ching Chen [7✉]

CD28 is required for T cell activation as well as the generation of CD4+Foxp3+ Treg. It is unclear, however, how CD28 costimulation affects the development of CD8+ T cell suppressive function. Here, by use of Hepa1.6.gp33 in vitro killing assay and B16.gp33 tumor mouse model we demonstrate that CD28 engagement during TCR ligation prevents CD8+ T cells from becoming suppressive. Interestingly, our results showed that ectonucleotidase CD73 expression on CD8+ T cells is upregulated in the absence of CD28 costimulation. In both murine and human tumor-bearing hosts, CD73 is upregulated on CD28−CD8+ T cells that infiltrate the solid tumor. UPLC-MS/MS analysis revealed that CD8+ T cells activation without CD28 costimulation produces elevated levels of adenosine and that CD73 mediates its production. Adenosine receptor antagonists block CD73-mediated suppression. Our data support the notion that CD28 costimulation inhibits CD73 upregulation and thereby prevents CD8+ T cells from becoming suppressive. This study uncovers a previously unidentified role for CD28 costimulation in CD8+ T cell activation and suggests that the CD28 costimulatory pathway can be a potential target for cancer immunotherapy.

[1] Department of Internal Medicine, National Taiwan University Hospital, Taipei, Taiwan. [2] Division of Colorectal Surgery, Department of Surgery, National Taiwan University Hospital and College of Medicine, Taipei, Taiwan. [3] Agricultural Biotechnology Research Center, Academia Sinica, Taipei, Taiwan. [4] Institute of Molecular Biology, Academia Sinica, Taipei, Taiwan. [5] Department of Chemistry, National Taiwan University, Taipei, Taiwan. [6] Department of Emergency Medicine, National Taiwan University Hospital, Taipei, Taiwan. [7] Department of Medical Research, National Taiwan University Hospital, Taipei, Taiwan. [8] Genomics Research Center, Academia Sinica, Taipei, Taiwan. [9] Department of Life Science, Tzu Chi University, Hualien, Taiwan. [10] Graduate Institute of Medical Sciences, National Defense Medical Center, Taipei, Taiwan. [11] Department of Obstetrics and Gynecology, National Taiwan University, College of Medicine, Taipei, Taiwan. [12] Graduate Institute of Immunology, National Taiwan University, College of Medicine, Taipei, Taiwan. ✉email: mdchensc@ntu.edu.tw

T cell activation requires two signals: T cell receptor (TCR) engagement with peptide/MHC complex sends the first signal and ligation of costimulatory molecules transduces the second one to fully activate T cells[1,2]. CD28 is the best characterized costimulatory molecule expressed by T cells. Upon TCR engagement, ligation of CD28 molecules results in T cell proliferation, increased IL-2 production, and survival[3,4]. CD28 engagement triggers phosphorylation of signaling molecules and activation of transcriptional programs to enhance TCR signaling[5]. It has been shown that TCR ligation in the absence of CD28 costimulation renders T cells anergic[6]. Interestingly, "anergic" T cells can act as specific suppressor cells[7,8], but its underlying mechanism is obscure.

CD28 signal is required for the generation and homeostasis of regulatory T cells (Treg cells). CD28-deficient mice are known to have reduced numbers of Treg cells[9], which predisposes those on the non-obese diabetic background to autoimmune disease[9]. Conditional knockout CD28 on CD4+Foxp3+ Treg cells (Cd28-ΔTreg mice) inhibits functional differentiation of CD4+ Treg cells[10]. Cd28-ΔTreg cells had lower expressions of CTLA4, PD-1, and CCR6 and retarded proliferative response to TCR stimulation even in the presence of B7-expressing antigen-presenting cells. In CD4+CD25−CD45RB$^{hi}$ cell-induced colitis in Rag$^{−/−}$ mice, adoptive transfer of Cd28-ΔTreg cells resulted in more severe colitis compared to mice receiving WT Treg cells, revealing that Cd28-ΔTreg cells failed to suppress colitis. Therefore, it appears that Treg cells lacking CD28 are functionally defective.

Recent studies have revealed that CD28 signaling modulates the regulatory function of CD8+ T cells and it is crucial to the development of spontaneous experimental autoimmune encephalomyelitis (EAE)[11]. Depletion of CD8+ T cells in CD28-deficient mice exacerbates EAE[11]. CD8-deficiency renders mice susceptible to EAE and it is reversed by adoptive transfer of CD28$^{−/−}$CD8+ T cells[12]. It was also shown that CD28−CD8+ T cells represent a unique subset of regulatory cells that induces a suppressive loop which may result in the induction of helper T cell unresponsiveness[12]. Accumulating evidence shows that the presence of tumor-infiltrating CD8+ T cells of the CD28− type is associated with advanced stages of cancer and poor patient survival[13,14]. These findings suggest that CD28−CD8+ T cells have an immunosuppressive role in regulating autoimmune disease, as well as in hampering anti-tumor responses. Whether CD28 signaling unlike that in CD4+ Treg cells curbs the generation and function of regulatory CD8+ T cells is an intriguing question to be addressed. We propose that in addition to the well-known CD28 role in enhancing TCR signaling and promoting T cell survival, it also directs activated CD8+ T cells away from a suppressive fate.

Foxp3 is known to regulate CD4+ Treg cell suppressive function[15] through cooperation with other transcription factors[16,17]. CD4+ Treg cells can secret inhibitory cytokine to exert their suppressive function[18]. Cell surface expression of both CD39 and CD73 on CD4 T cells is also linked to their suppressive function[19], which is mediated by CD39 and CD73 through conversion of ATP/ADP to AMP and AMP to adenosine, respectively[20]. Extracellular adenosine generated by CD4+ Treg cells acts to block T cell activation and proliferation and inhibits effector T cell cytotoxicity and cytokine-producing functions[21,22]. It will be of interest to determine whether CD8+ suppressive T cells exert immunosuppressive function by the same mechanism as CD4+ Treg cells.

Here, we examined the role of CD28 costimulation in regulating the suppressive function of CD8+ T cells. We showed that CD28 costimulation interdicts the inhibitory function of TCR-stimulated CD8+ T cells through abrogating CD73 expression. Our work substantiates a previously unidentified role of CD28 costimulation in CD8+ T cell activation.

## Results

### CD28 costimulation prevents CD8+ T cells from becoming suppressive.

To investigate the role of CD28 signaling in the development of CD8+ T cell suppressive function, CD8+ T cells from CD28$^{+/+}$ (CD28WT) and CD28$^{−/−}$ (CD28KO) mice were stimulated with anti-CD3 plus anti-CD28 (anti-CD3/28) antibodies for 3 days. The cells were harvested and labeled with CFSE for further assessment of their suppression on CD8+ effector T cells (CD8+ T$_{eff}$). CD8+ effector T cells were obtained by stimulation of CD28$^{+/+}$CD8+ T cells from P14 TCR transgenic mice with specific antigenic LCMV gp33-41 M2 peptide which recognizes LCMV gp33 in the context of H-2D$^b$ on B16.gp33 melanoma. P14 CD8+ T$_{eff}$ (CFSE-negative) were co-cultured with CFSE-labeled CD28WTCD8+ or CD28KOCD8+ T cells and the effects of activated CD28WTCD8+ and CD28KOCD8+ T cells on the ability of CD8+ T$_{eff}$ cells to kill tumor were assessed (Fig. 1a). Hepa 1-6.gp33 tumor cells bearing H-2D/K$^b$ MHC class I molecules were used as CTL targets. Results of this in vitro assay showed that compared to activated-CD28KOCD8+ T cells-pretreated P14 CD8+ T$_{eff}$ cells, activated CD28WTCD8+ T cells-pretreated P14 CD8+ T$_{eff}$ cells had significantly higher ability to kill Hepa 1-6.gp33 targets (Fig. 1b). Since the percentage of tumor-killing was comparable between pretreatment of P14 CD8+ T$_{eff}$ cells with activated CD28WTCD8+ T cells and control P14 CD8+ T$_{eff}$ cells alone, the cytolytic activity of T$_{eff}$ cells co-cultured with activated-CD28WTCD8+ T cells was superior to T$_{eff}$ cells co-cultured with activated CD28KOCD8+ T cells and did not the result from enhanced tumor-killing by the former.

The effect of activated CD28WTCD8+ T cells and CD28KOCD8+ T cells on the tumor-killing ability of CD8+ T$_{eff}$ cells was further tested in B16.gp33 melanoma-bearing C57BL/6 (B6) mice. P14 CD8+ T$_{eff}$ cells that had been pretreated with either activated CD28WTCD8+ T cells or CD28KOCD8+ T cells were adoptively transferred to tumor-bearing mice (Fig. 1a). Mice that received activated CD28KOCD8+ T cells-pretreated P14 CD8+ T$_{eff}$ cells developed tumors that were larger in size than those receiving P14 CD8+ T$_{eff}$ cells that were pretreated with activated CD28WTCD8+ T cells (Fig. 1c). These results suggest that CD28 costimulation promotes anti-tumor CD8+ T cell response by preventing the induction of suppressive function.

### CD28 costimulation regulates the release of soluble inhibitory factor(s) by TCR-activated CD8+ T cells.

To study whether CD28KOCD8+ T cells suppress effector cell anti-tumor activity through direct contact or through secreting soluble mediator(s), we added supernatants from anti-CD3/28-activated CD28WTCD8+ (Sup-CD28WTCD8+) and CD28KOCD8+ (Sup-CD28KOCD8+) T cell cultures to P14 CD8+ T$_{eff}$ separately and examined P14 CD8+ T$_{eff}$ anti-tumor activity (Fig. 1d). Results in Fig. 1e showed that the ability of Sup-CD28WTCD8+-treated P14 CD8+ T$_{eff}$ cells to kill Hepa 1-6.gp33 target was greater than P14 CD8+ T$_{eff}$ cells treated with Sup-CD28KOCD8+ at all E:T ratios tested. Results of adoptive transfer experiments showed that Sup-CD28WTCD8+ treated-P14 CD8+ T$_{eff}$ cells were more efficient than Sup-CD28KOCD8+-treated P14 CD8+ T$_{eff}$ cells in controlling B16.gp33 melanoma growth in mice (Fig. 1f). Coinciding with slower tumor growth, tumor-bearing mice receiving Sup-CD28WTCD8+ treated-P14 CD8+ T$_{eff}$ cells survived significantly longer than those receiving Sup-CD28KOCD8+-treated P14 CD8+ T$_{eff}$ cells (Fig. 1g).

Transwell experiments (Fig. 1h) further showed that soluble mediator(s) released by CD28KOCD8+ T cells significantly reduced the ability of P14 CD8+ T$_{eff}$ to kill Hepa 1-6.gp33 targets in vitro (Fig. 1i, j). P14 CD8+ T$_{eff}$ cells exposed through transwells

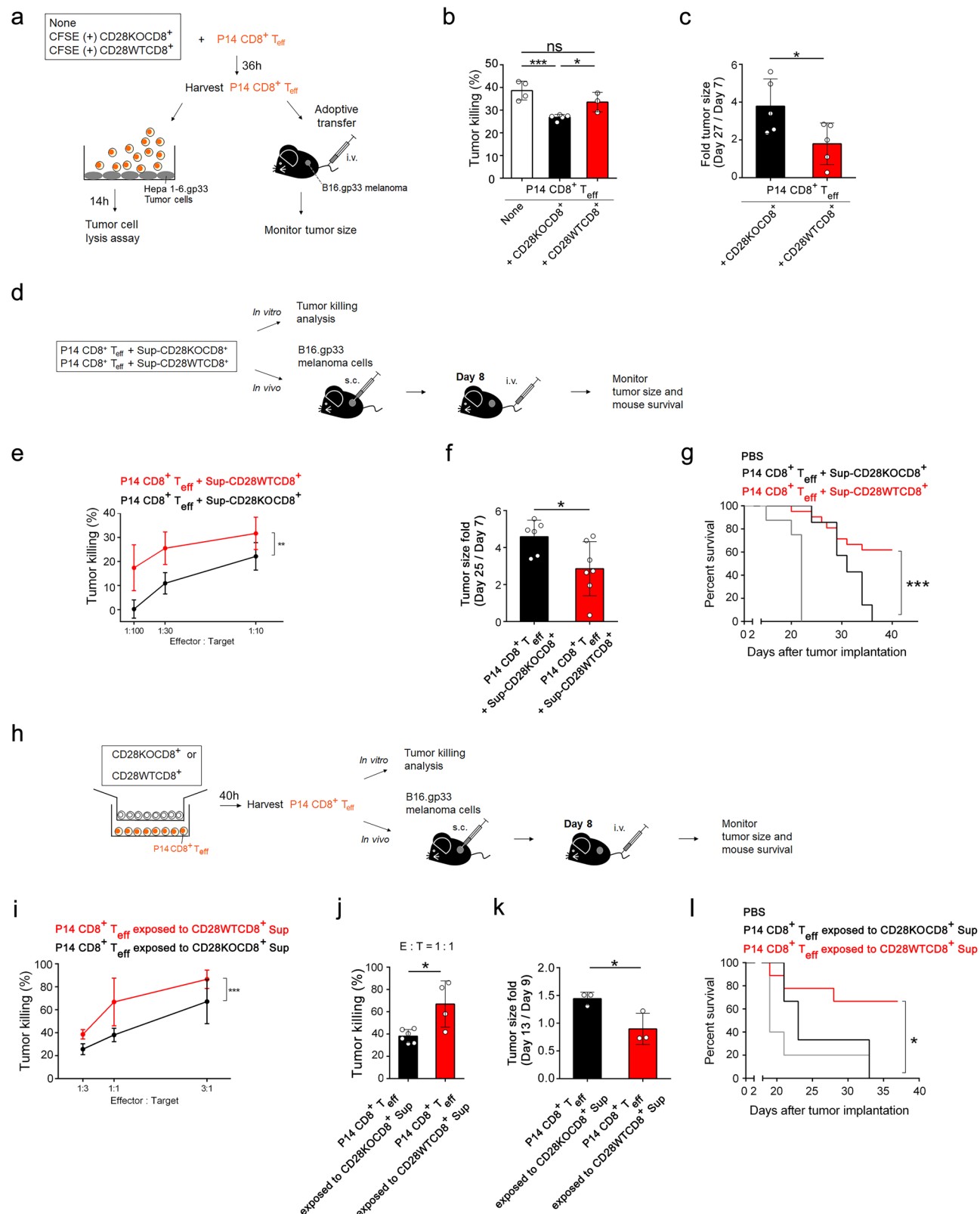

to soluble mediators released by CD28KOCD8+ T cell were less effective than those exposed to that by CD28WTCD8+ in controlling tumor growth (Fig. 1k) and mice receiving P14 CD8+ $T_{eff}$ cells exposed to soluble mediators from CD28KOCD8+ T cells had significantly poorer survival than those exposed to that from CD28WTCD8+ (Fig. 1l). Taken together, CD28 costimulation downregulates the release of soluble factor(s) by TCR-

activated CD8+ T cells that suppresses the anti-tumor cytolytic function of CD8+ effector T cells.

**Stimulation through TCR in the absence of CD28 confers CD8+ T cells the ability to suppress effector T cell cytokine production through soluble inhibitory mediators other than IL-6, IL-10, and IL-17.** To examine whether CD8+ T cells activated by TCR

**Fig. 1 CD28 costimulation prevents CD8$^+$ T cells from becoming suppressive by blocking the release of soluble inhibitory factor(s). a** Experimental design for **b**, **c**. CFSE-labeled anti-CD3/28-stimulated CD28WTCD8$^+$ or CD28KOCD8$^+$ T cells were added to P14 CD8$^+$ T$_{eff}$ culture (CFSE$^-$) with a ratio of CFSE$^+$:CFSE$^-$ of 1:3. **b** CD28KOCD8$^+$ ($n = 5$, black)- or CD28WTCD8$^+$ ($n = 3$, red)-pretreated P14 CD28$^{+/+}$ T$_{eff}$ were added to Hepa 1-6.gp33 cells at an E:T ratio of 1:3. *Blank*, P14 CD8$^+$ T$_{eff}$ without treatment ($n = 4$). **c** Fold increase of tumor size: 3.8 ± 0.65 vs. 1.8 ± 0.49, day 27/day 7 after tumor implantation. *Black*, receiving CD28KOCD8$^+$ T cells-pretreated P14 CD8$^+$ T$_{eff}$. *Red*, receiving CD28WTCD8$^+$ T cells-pretreated P14 CD8$^+$ T$_{eff}$. $n = 5$. **d** Experimental design for **e–g**. **e** Sup-CD28KOCD8$^+$ ($n = 3$, black)- and Sup-CD28WTCD8$^+$ ($n = 4$ at E:T ratio of 1:100 and 1:30, $n = 3$ at E:T ratio of 1:10, red)-treated P14 CD28$^{+/+}$ T$_{eff}$ were added to Hepa 1-6.gp33 cells at the indicated E:T ratios. Data were analyzed by a general linear model. **f** Fold increase of tumor size. *Black*, receiving Sup-CD28KOCD8$^+$-treated P14 CD8$^+$ T$_{eff}$ ($n = 6$). *Red*, receiving Sup-CD28WTCD8$^+$-treated P14 CD8$^+$ T$_{eff}$ ($n = 7$). **g** Survival of tumor-bearing mice until 40 days after tumor implantation. *Gray*, injection with PBS ($n = 8$). Receiving Sup-CD28KOCD8$^+$ ($n = 7$, black)- or Sup-CD28WTCD8$^+$ ($n = 21$, red)-treated P14 CD8$^+$ T$_{eff}$. **h** Experimental design for **i–k**. Transwell experiment: Insert: activated CD28KOCD8$^+$ or CD28WTCD8$^+$ T cells; lower chamber: P14 CD8$^+$ T$_{eff}$ cells; Insert:lower chamber cell number ratio = 3:1. **i** Percent killing of Hepa 1-6.gp33 cells by P14 CD8$^+$ T$_{eff}$ cells at the indicated E:T ratios. *Black*, exposure to CD28KOCD8$^+$ T cell culture supernatant ($n = 6$); *Red*, exposure to CD28WTCD8$^+$ T cell culture supernatant ($n = 4$ at E:T ratio of 1:3 and 1:1; $n = 5$ at E:T ratio of 3:1). Data were analyzed by a general linear model. **j** Further analysis of data in **i**. Tumor killing at the E:T ratio of 1:1 (P14 CD8$^+$ T$_{eff}$ exposed to CD28KOCD8$^+$ Sup, $n = 6$; P14 CD8$^+$ T$_{eff}$ exposed to CD28KOCD8$^+$ Sup, $n = 4$). **k** Fold increase of tumor size. Transfer of P14 CD8$^+$ T$_{eff}$ cells exposed to CD28KOCD8$^+$ T cell culture supernatants (*black*) or those exposed to CD28WTCD8$^+$ T cell culture supernatants (*red*). ($n = 3$) **l** Animal survival until 37 days after tumor implantation. Injection with PBS ($n = 5$, gray). Transfer of P14 CD8$^+$ T$_{eff}$ cells exposed to CD28KOCD8$^+$ T cell culture supernatants ($n = 3$, black) or to CD28WTCD8$^+$ T cell culture supernatants ($n = 9$, red). s.c., subcutaneously; i.v., intravenously. Statistical evaluations were performed using the Student's *t*-test with data expressed as the mean ± standard error of the mean (**b**, **c**, **f**, **j**, **k**). The ability of CD8$^+$ T cells between two groups to in vitro kill tumor at different E:T ratio was analyzed by a general linear model (**e**, **i**). The survival difference between groups was analyzed by log-rank test (**g**, **l**). *$p < 0.05$, **$p < 0.01$, ***$p < 0.001$; ns, not statistically significant.

without CD28 costimulation also suppress effector T cell cytokine production, we co-cultured CD8$^+$ T$_{eff}$ with CD28KOCD8$^+$ or CD28WTCD8$^+$ T cells that had been stimulated by anti-CD3/28 antibodies. While high percentages of CD8$^+$ T$_{eff}$ cells pre-cultured with activated CD28WTCD8$^+$ T cells produced TNF, IFN-γ, perforin, and granzyme B, pre-culture with activated CD28KOCD8$^+$ T cells reduced the fraction of cells TNF-, IFN-γ-, and granzyme B-, but not perforin-producing cells (Fig. 2a). Decreased mean fluorescence intensity (MFI) of intracellular staining revealed the reduced ability of CD8$^+$ T$_{eff}$ cells pre-cultured with CD28KOCD8$^+$ T cells to produce TNF and IFN-γ compared to those pre-cultured with activated CD28WTCD8$^+$ T cells (Fig. 2b). Supernatants from anti-CD3- (anti-CD3 Sup) and anti-CD3/28- (anti-CD3/28 Sup) activated CD28WTCD8$^+$ T cell cultures had a similar effect as CD28KOCD8$^+$ T cells and CD28WTCD8$^+$ T cells, respectively, on CD8$^+$ T$_{eff}$ cytokine production except that anti-CD3 Sup significantly reduced the ability of CD8$^+$ T$_{eff}$ granzyme B production (Fig. 2c). Compared to anti-CD3/28 Sup, anti-CD3 Sup also reduced the ability of CD4$^+$ T cells to produce IL-2, TNF, and IFN-γ (Supplementary Fig. 1a) but only had a marginal effect on suppressing CD4$^+$ T cell proliferation (Supplementary Fig. 1b).

IL-10, IL-6, and IL-17 are known to regulate T cell immune response. We found that stimulation of CD28WTCD8$^+$ T cells with either anti-CD3 or anti-CD3/28 antibodies induced IL-10, IL-6, and IL-17 production, and stimulation with anti-CD3 antibodies alone did not increase the levels of these cytokines (Fig. 2d). These results together suggest that CD8$^+$ T cells acquire suppressive function when stimulated by TCR without CD28 costimulation and that the suppressive CD8$^+$ T cells so generated inhibit CD8$^+$ T$_{eff}$ cytolytic activity and CD4$^+$ T cell cytokine production through releasing soluble inhibitory mediator(s) other than IL-10, IL-6, and IL-17.

**CD28 costimulatory signaling inhibits CD73 expression on CD8$^+$ T cells.** To investigate the role of CD28 in regulating the expression of proteins involved in the suppressive function of CD8$^+$ T cells, we examined naïve CD28WTCD8$^+$ and CD28KOCD8$^+$ T cells activated by anti-CD3/28 antibody for the signature protein expression that defines CD4$^+$ Treg cells. Among the 15 Treg signature proteins, a protein that is associated with suppressive function, CD73, was upregulated in activated CD28KOCD8$^+$ T cells as compared to activated CD28WTCD8$^+$ T cells (Fig. 3). Interestingly, there was no difference in Foxp3

expression between CD28KOCD8$^+$ T cells and CD28WTCD8$^+$ T cells stimulated with anti-CD3/28 antibodies (Fig. 3). These results demonstrate that CD28 signaling inhibits TCR-induced CD73 expression in CD8$^+$ T cells but is without effect on Foxp3 expression.

CD73 protein upregulation was confirmed in CD28WTCD8$^+$ T cells stimulated with anti-CD3 antibody (Fig. 4a, b) and in CD28KOCD8$^+$ T cells stimulated with anti-CD3/28 antibodies (Fig. 4c, d and Supplementary Fig. 5). Interestingly, CD39 that is known to coordinately expressed with CD73 in generating suppressive loops[19,20,23,24] was not upregulated in CD28KOCD8$^+$ T cells stimulated with anti-CD3/28 antibodies (Fig. 4c), P14 CD28KOCD8$^+$ T cells stimulated with M2 peptide (Fig. 4e) or in CD28$^{lo}$CD8$^+$ T cells stimulated with anti-CD3/28 antibodies (Fig. 4f). Inhibition of CD28 signaling by PI3K inhibitor upregulated CD73 expression in CD28WTCD8$^+$ T cells stimulated with anti-CD3/28 antibodies to a level comparable to that in cells stimulated with anti-CD3 antibody alone (Fig. 4g, h), both of which were significantly higher than that in cells treated with anti-CD3/28 antibodies. Subjecting supernatants to UPLC-MS/MS analysis (Supplementary Fig. 2a), the level of adenosine detected in the 2-h culture supernatants of anti-CD3-stimulated CD28WTCD8$^+$ T cells (0.19 ± 0.011 μM) was significantly higher than that of anti-CD3/28-stimulated CD28WTCD8$^+$ T cells (0.13 ± 0.014 μM) ($p = 0.0048$) and that no adenosine was detected in CD73-knockout, CD28-sufficient CD8$^+$ (CD28WTCD73KOCD8$^+$) T cell culture supernatants (Fig. 4i and Supplementary Fig. 2b). Consistently, residual AMP in the supernatant of anti-CD3-stimulated CD28WTCD8$^+$ T cells (0.03 ± 0.004 μM) was lower than that in anti-CD3/28-stimulated CD28WTCD8$^+$ T cells (0.06 ± 0.006 μM) ($p = 0.0096$) (Supplementary Fig. 2c). In addition, the levels of AMP were comparable in culture supernatants of CD73KOCD8$^+$ T cells stimulated with either anti-CD3 or anti-CD3/CD28 antibodies (Supplementary Fig. 2c).

ATP released by dying cells can be rapidly converted by CD73 to AMP and further to adenosine. We added saturation levels of AMP$_{13C,15N}$ isotope (37.5 μM) to anti-CD3-stimulated and anti-CD3/CD28-stimulated CD73-sufficient and CD73-knockout CD28WTCD8$^+$ T cell cultures with or without the addition of CD73 inhibitor and measured AMP$_{13C,15N}$ conversion to adenosine. While no AMP to adenosine conversion occurred in CD73-knockout cell cultures, the level of adenosine was significantly higher in CD73-sufficient cell cultures stimulated

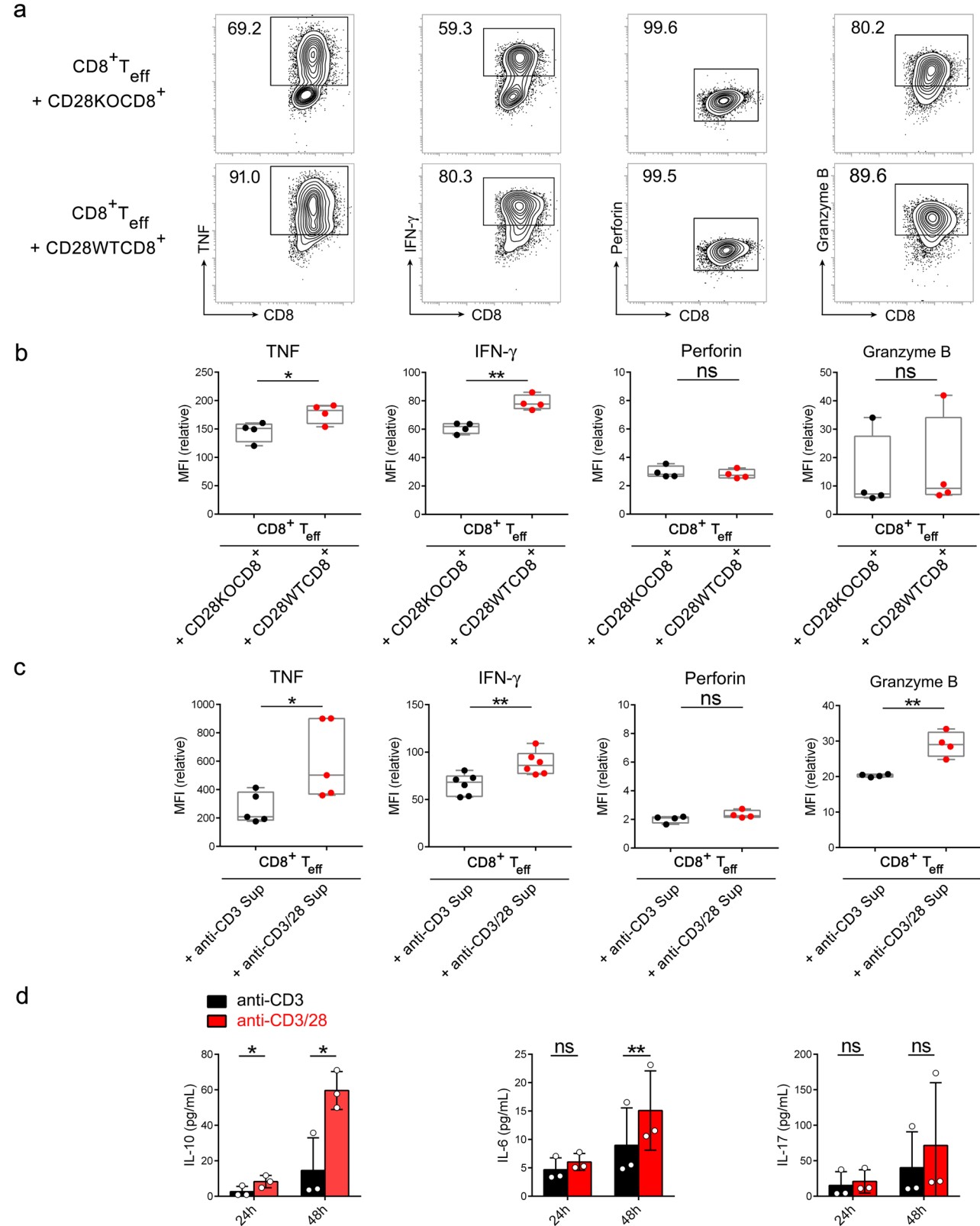

with anti-CD3 antibody ($13.70 \pm 1.149\,\mu M$) than in cultures stimulated with anti-CD3/CD28 antibodies ($6.27 \pm 0.716\,\mu M$), and CD73 inhibitor almost completely inhibited AMP conversion to adenosine (Fig. 4j and Supplementary Fig. 2d). The levels of residual $AMP_{13C,15N}$ in anti-CD3-stimulated ($23.51 \pm 1.232\,\mu M$, conversion rate 37.3%) and anti-CD3/CD28-stimulated ($28.79 \pm 1.277\,\mu M$, conversion rate 23.4%) cultures indicate that

CD28 signaling resulted in less efficient conversion of AMP to adenosine (Supplementary Fig. 2e). These results together demonstrate that the conversion of AMP to adenosine is mediated by the CD73 activity of viable cells. CD28 signaling alongside TCR stimulation, therefore, prevents CD73 expression, and that CD73-mediated conversion of AMP into adenosine renders CD8$^+$ T cell suppressive.

**Fig. 2 CD8$^+$ T cells stimulated via TCR without CD28 costimulation suppress effector T cell cytokine production. a** Naïve CD8$^+$ T cells from CD28KO or CD28WT mice were stimulated by anti-CD3/28 antibodies. The cells were labeled with CFSE on day 3 after stimulation and added to CFSE$^-$CD8$^+$ T$_{eff}$ culture with a ratio of CFSE$^+$: CFSE$^-$ of 1:3. On 36 h after co-culture, CFSE$^-$ cells in each group were sorted and restimulated with PMA and ionomycin for 6 h. The production of TNF, IFN-γ, perforin, and granzyme B by CD8$^+$ T$_{eff}$ cells was detected by intracellular staining and analyzed by flow cytometry. Numbers on the upper left corner indicate the percentage of intracellular protein-positive cells in the total CD8$^+$ T cell population. The data presented are representative of one of the three independent experiments. **b** Relative medium fluorescence intensity (MFI) of each intracellular protein in CD8$^+$ T$_{eff}$ cell after co-culture with CD28KOCD8$^+$ (black) or CD28WTCD8$^+$ T cells (red). ($n = 4$). **c** Day-3 culture supernatants collected from anti-CD3- (black) or anti-CD3/28- (red) stimulated CD28WTCD8$^+$ T cells were added to CD8$^+$ T$_{eff}$ cells and cultured for 24 h. The production of intracellular proteins by CD8$^+$ T$_{eff}$ cells was assessed by relative MFI (TNF, $n = 5$; IFN-γ, $n = 6$; perforin, $n = 4$; granzyme B, $n = 4$). **d** Culture supernatants from anti-CD3/28 (red)-stimulated and anti-CD3 (black)-stimulated CD28WTCD8$^+$ T cells were collected on 24 and 48 h after stimulation. IL-10, IL-6, and IL-17 concentrations in the supernatants were quantified by Cytometric Bead Array. ($n = 3$). Statistical evaluations were performed using the Student's t-test with data expressed as the mean ± standard error of the mean (**b–d**). *$p < 0.05$, **$p < 0.01$; ns, not statistically significant.

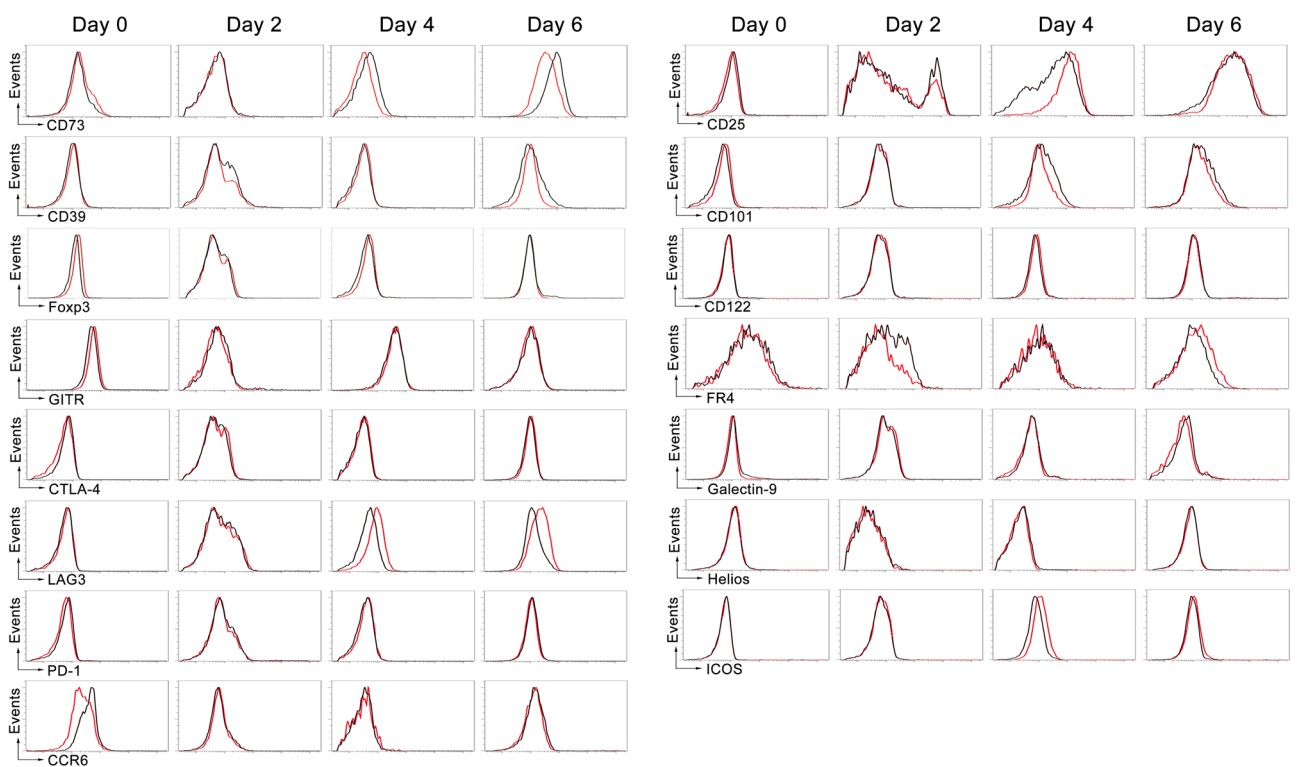

**Fig. 3 Regulatory signature of CD8$^+$ T cells stimulated via TCR without CD28 costimulation.** Naïve CD8$^+$ T cells ($3 \times 10^6$/mL) from spleens of CD28KO (black) and CD28WT (red) mice were sorted and stimulated with anti-CD3/28 antibodies. On days 0, 2, 4, and 6 after stimulation, cells were harvested and subjected to flow cytometric analysis for the expressions of CD73, CD39, Foxp3, GITR, CTLA-4, LAG3, PD-1, CCR6, CD25, CD101, CD122, FR4, Gelactin-9, Helios and ICOS. The data presented are representative of one of the three independent experiments.

**CD73 expression on CD8$^+$ T cells in tumor-bearing hosts negatively correlates with CD28 expression.** We next explored the causal relationship between CD73 and CD28 expressions on CD8$^+$ T cells in tumor-bearing mice. Mice were inoculated with CD73-negative B16.gp33 tumor cells (Supplementary Fig. 3) and CD73 expression on CD28$^-$CD8$^+$ and CD28$^+$CD8$^+$ T cells in tumor-draining lymph nodes and tumor were analyzed. Results showed that the levels of CD73 were significantly higher on CD28$^-$CD8$^+$ T cells than on CD28$^+$CD8$^+$ T cells isolated from the draining lymph nodes on day 8 (Fig. 5a) and from tumors on day 20 (Fig. 5b) after tumor implantation.

To address possible clinical relevance, human CD8$^+$ T cells in PBL, as well as tumor microenvironment from colon cancer patients, were studied for correlated CD73 vs. CD28 expression. We found that while CD73 expression in tumor-infiltrating CD28$^+$CD8$^+$ T cells was slightly upregulated compared to CD28$^+$CD8$^+$ T cells in the peripheral blood (MFI TIL/PBMC ratio = 1.61 ± 0.226), CD73 expression in CD28$^-$CD8$^+$ T cells isolated from tumor had significantly higher levels of CD73 than CD28$^-$CD8$^+$ T cells in

the circulation (MFI TIL/PBMC ratio = 4.44 ± 0.822) ($p < 0.0001$) (Fig. 5c). We further performed paired analysis of CD73 expression on CD28$^+$CD8$^+$ and CD28$^-$CD8$^+$ T cells in peripheral blood and in tumors from the same patient. Results in Fig. 5d showed that there was significantly higher CD73 upregulation on CD28$^-$CD8$^+$ T cells than that on CD28$^+$CD8$^+$ when they were both in the tumor microenvironment (Fig. 5d). These results support the notion that CD73 expression on CD8$^+$ T cells is associated with the absence of CD28 costimulation. In light of the fact that adenosine generated via CD73 suppresses anti-tumor effector T cell function, these findings suggest that providing sufficient CD28 signaling in the tumor microenvironment can inhibit CD73 expression on CD8$^+$ T cells thereby promotes anti-tumor immunity.

**Activated CD28KOCD8$^+$ T cells suppress effector T cell function through CD73-mediated adenosine production.** Pre-culture with CD28KOCD8$^+$ T cells reduced TNF and IFN-γ production by P14 CD8$^+$ T cells (Fig. 6a). On the other hand,

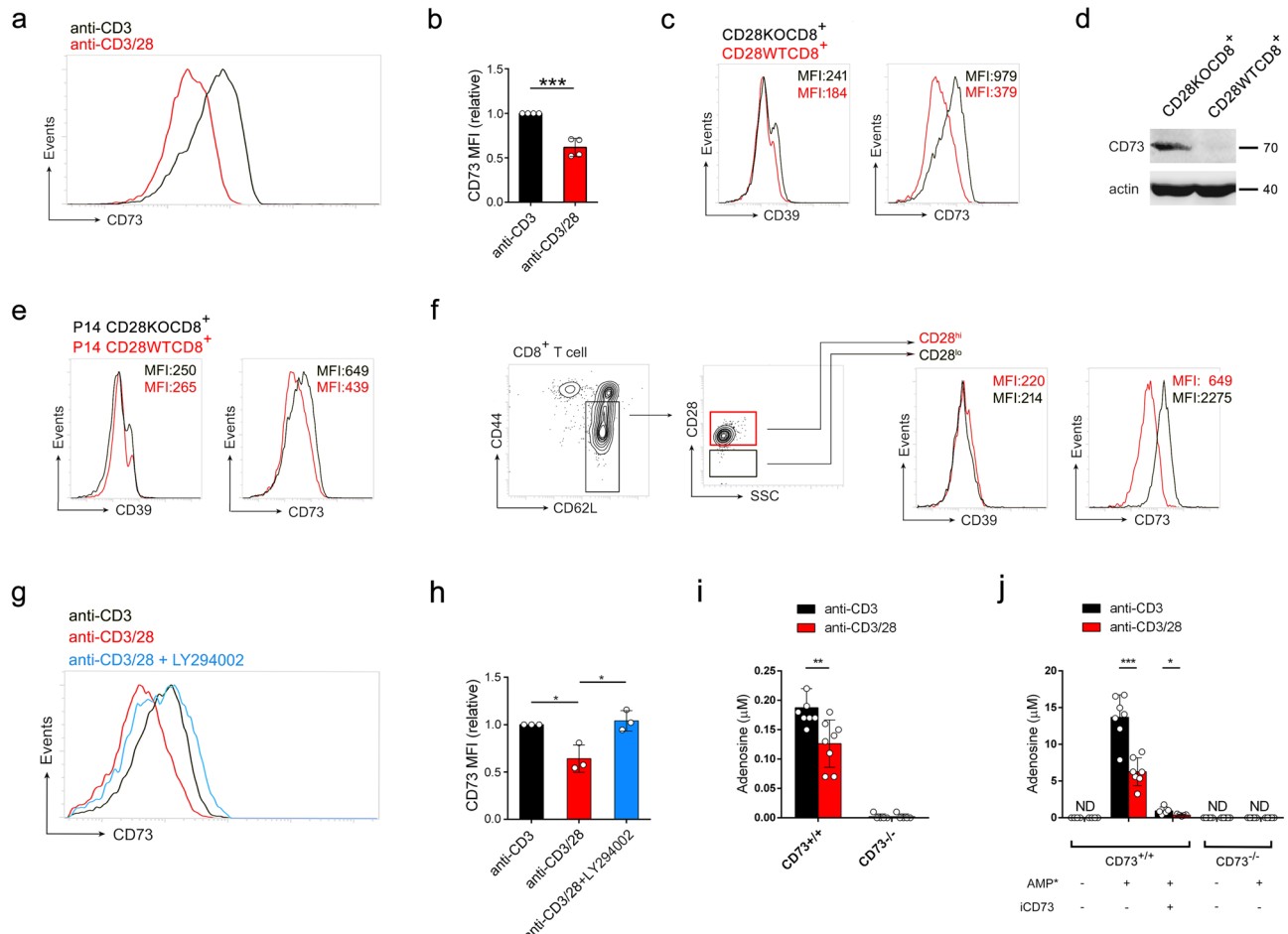

**Fig. 4 CD28 costimulation inhibits CD73 upregulation on TCR-activated CD8$^+$ T cells. a** Naïve CD28WTCD8$^+$ T cells were sorted and stimulated with anti-CD3 antibody in the presence (red) or absence (black) of anti-CD28 antibody. Histogram of CD73 expression on CD8$^+$ T cells on day 5 after stimulation. The data presented are representative of one of the four independent experiments. **b** As **a**, relative MFI of CD73 expression. ($n = 4$). MFI of CD73 in cells stimulated with anti-CD3 antibody (black) was taken as 1.0. MFI of CD73 in cells stimulated with anti-CD3/28 antibodies (red) was normalized against that in cells stimulated with anti-CD3 antibody (relative MFI). **c** Naïve CD8$^+$ T cells from CD28KO (black) and CD28WT (red) mice were stimulated with anti-CD3/28 antibodies for 5 days and the cells were subjected to flow cytometric analysis for CD39 and CD73 expressions. The data presented are representative of one of the three independent experiments. **d** Naïve CD8$^+$ T cells from CD28KO and CD28WT mice were stimulated with anti-CD3/28 antibodies for 5 days and the cell lysate was subjected to Western blot analysis for CD73 expression. The blot represents three independent experiments. **e** Naïve CD8$^+$ T cells from P14 CD28KO (black) and P14 CD28WT (red) mice were stimulated with M2 peptide-pulsed B cell blasts. On day 5 after stimulation, cells were subjected to flow cytometric analysis for CD39 and CD73 expressions. The data presented are representative of one of the three independent experiments. **f** CD28$^{hi}$ (red) and CD28$^{lo}$ (black) populations sorted from naïve CD28WTCD8$^+$ T cells were stimulated with anti-CD3/28 for 5 days and subjected to flow cytometric analysis of CD39 and CD73 expressions. The data presented are representative of one of the three independent experiments. **g** Naïve CD28WTCD8$^+$ T cells were stimulated by anti-CD3 (*black*) or by anti-CD3/28 (*red*) antibodies in the presence (blue) or absence of LY294002 (3 µM). Histogram of CD73 expression on CD8$^+$ T cells on day 5 after stimulation. The data presented are representative of one of the three independent experiments. **h** As **g**, relative MFI with that stimulated with anti-CD3 antibody as 1.0 (black). Anti-CD3/28, *red*. Anti-CD3/28 + LY294002, *blue*. ($n = 3$). **i** Culture supernatants of anti-CD3 (black)- and anti-CD3/28 (red)-stimulated CD8$^+$ T cells from CD73$^{+/+}$ ($n = 8$) and CD73$^{-/-}$ ($n = 5$) mice were collected after 2-h incubation in serum-free IL-2-containing DMEM and were subjected to UPLC-MS/MS analysis to quantify adenosine. **j** The capacity of CD8$^+$ T cells to degrade AMP into adenosine was analyzed after 2-h incubation with AMP$_{13C,15N}$ isotope (AMP*, 37.5 µM). Culture supernatants of anti-CD3 (black)- and anti-CD3/28 (red) antibodies-stimulated CD8$^+$ T cells from CD73$^{+/+}$ ($n = 7$) and CD73$^{-/-}$ ($n = 5$) mice were collected after 2-h incubation in serum-free IL-2-containing DMEM and were subjected to UPLC-MS/MS analysis to quantify adenosine$_{13C,15N}$. Neither pre-incubation of cells with CD73 inhibitor iCD73 ($n = 5$) before the addition of AMP$_{13C,15N}$ nor stimulated CD73$^{-/-}$ CD8$^+$ T cells produced significant amounts of adenosine$_{13C,15N}$, confirming the role of CD73 in the production of adenosine. Statistical evaluations were performed using the Student's *t*-test with data expressed as the mean ± standard error of the mean (**b**, **h**, **i**, **j**). *$p < 0.05$, **$p < 0.01$, ***$p < 0.001$.

pre-cultured with CD73 inhibitor-pretreated activated CD28KOCD8$^+$ T cells or treatment of P14 CD8$^+$ T cells with adenosine A$_{2A}$ receptor antagonist (ZM241385) and A$_{2B}$ receptor antagonist (MRS1754) resulted in increased TNF and IFN-γ production to levels comparable to those pre-cultured with CD28WTCD8$^+$ T cells (Fig. 6a). We next explored the effect of adenosine produced by activated CD28KOCD8$^+$ T cells on the

cytolytic activity of P14 CD8$^+$ T$_{eff}$ cells against Hepa 1-6.gp33 tumor cells. Pretreatment with CD73 inhibitor significantly diminished the suppressive effect of activated CD28KOCD8$^+$ T cell supernatant on P14 CD8$^+$ T$_{eff}$ cells (Fig. 6b). At a 1:10 E:T ratio, supernatants from activated CD28KOCD8$^+$ T cell culture abolished the cytolytic activity of P14 CD8$^+$ T$_{eff}$ cells against tumor targets compared to supernatant from activated

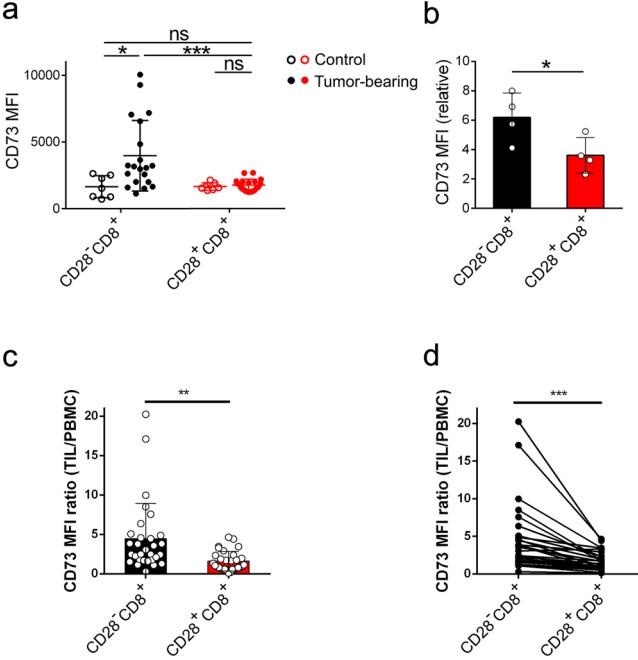

**Fig. 5 CD28 expression on CD8$^+$ T cells in the tumor microenvironment negatively correlates with CD73 expression. a** Mice were subcutaneously injected with B16.gp33 melanoma cells ($2 \times 10^6$ cells per mouse). Eight days later, tumor-draining lymph nodes were harvested. Total lymph node cells were subjected to flow cytometric analysis for CD73 expression on CD28$^-$CD8$^+$ (solid black) and CD28$^+$CD8$^+$ (solid red) T cells ($n = 20$, each). Lymph nodes harvested from tumor-free mice (controls) were subjected to flow cytometric analysis for CD73 expression on CD28$^-$CD8$^+$ (empty black) and CD28$^+$CD8$^+$ (empty red) T cells ($n = 7$, each). **b** As **a**, on day 20 after B16.gp33 implantation, tumors were dissected and dissociated into single cells. Single-cell suspensions were subjected to flow cytometric analysis for CD73 expression on CD28$^-$CD8$^+$ (black) and CD28$^+$CD8$^+$ (red) T cells ($n = 4$). Relative MFI of CD73 on CD28$^-$CD8$^+$ and CD28$^+$CD8$^+$T cells was compared against isotype control. **c** CD73 expressions on CD28$^-$CD8$^+$ ($n = 30$, black) and CD28$^+$CD8$^+$ ($n = 30$, red) CD8$^+$ T cells in TILs were compared to their respective counterparts in PBMC (TIL/PBMC) of colon cancer. **d** CD73 expressions on CD28$^-$CD8$^+$ and CD28$^+$CD8$^+$ CD8$^+$ T cells in TILs were compared to their respective counterparts in PBMC from the same individual. CD73 MFI from TILs and PBMC was analyzed by paired student $t$-test ($n = 30$). Statistical evaluations were performed using the Student's $t$-test with data expressed as the mean $\pm$ standard error of the mean (**a**–**d**). \*$p < 0.05$, \*\*$p < 0.01$, \*\*\*$p < 0.001$; ns, not statistically significant.

CD28WTCD8$^+$ T cell culture (Fig. 6c). At the same E:T ratio, supernatants from CD73 inhibitor-pretreated activated CD28KOCD8$^+$ T cells had no suppressive effect on the ability of P14 CD8$^+$ T$_{eff}$ to kill tumor cells. To address whether adenosine mediates the suppressive function of activated CD28KOCD8$^+$ T cells on CD8$^+$ T$_{eff}$ anti-tumor activity in vivo, B16.gp33 tumor-bearing mice were given sorted CFSE$^-$P14 CD8$^+$ T$_{eff}$ cells pre-cultured with CFSE$^+$CD28KOCD8$^+$ T cells that were treated with or without inhibitors. A higher fraction of tumor-bearing mice survived after they were adoptively transferred with CFSE-negative P14 CD8$^+$ T$_{eff}$ cells pre-cultured with activated CD28WTCD8$^+$ T cells (71.7% survival, group I), compared to those that were adoptively transferred with P14 CD8$^+$ T$_{eff}$ cells pre-cultured with activated CD28KOCD8$^+$ T cells (51.4% survival, group II) (Fig. 6d). Transfer of P14 CD8$^+$ T$_{eff}$ pre-cultured with CD73 inhibitor-pretreated activated CD28KOCD8$^+$ T cells (86.2% survival, group III) resulted in significantly increased survival of tumor-bearing mice compared to mice receiving P14

CD8$^+$ T$_{eff}$ cells pre-cultured with activated CD28KOCD8$^+$ T cells (group II) ($p < 0.01$). In addition, transfer of P14 CD8$^+$ T$_{eff}$ that had been pre-cultured with activated CD28KOCD8$^+$ T cells in the presence of ZM241385 (79.5% survival, group IV) also significantly improved the survival of tumor-bearing mice compared to mice in group II ($p < 0.01$). These data together show that CD28 costimulation inhibits the expression of CD73, with concomitant production of adenosine, which in turn mediates suppression of the cytolytic activity of CD8$^+$ T$_{eff}$ cell against tumor cells.

## Discussion

To fully activate T cells, CD28 costimulation accompanying TCR engagement protects T cells from apoptosis and anergy. As a result, T cells undergo clonal expansion and functional differentiation[25,26]. In this study, we uncovered a function of CD28 costimulation for CD8$^+$ T cell activation. CD28 costimulation blocks the expression of CD73 and prevents CD8$^+$ T cells from becoming suppressive. CD28 signaling inhibits CD73-mediated adenosine production and enables CD8$^+$ T cells to acquire anti-tumor effector function. Our results define an immunoregulatory role of CD28 costimulation in CD8$^+$ T cells.

CD4$^+$CD25$^+$Foxp3$^+$ Treg cells are the most well-characterized population of regulatory T cells. CD4$^+$CD25$^+$Foxp3$^+$ Treg cells require CD28 signaling for their generation[27,28]. It is also reported that T cell activation without CD28 costimulation renders T cells anergic, a hyporesponsive state where T cells are at growth arrest and unable to produce cytokines and to carry out effector function[6]. CD73$^{hi}$ suppressive CD8$^+$ T cells we report in this study are an unidentified population. They are generated by TCR stimulation without CD28 costimulation. They are capable of proliferation (Supplementary Fig. 4a) and production of IL-2, TNF, IFN-$\gamma$, and granzyme B (Supplementary Fig. 4b). The amount of IL-2, IFN-$\gamma$, and granzyme B they produce is at a level comparable to their counterparts that receive CD28 costimulation (Supplementary Fig. 4c), indicating that these CD73$^{hi}$ suppressive CD8$^+$ T cells still retain their cytotoxic potential and the ability to produce cytokine. It appears that the requirement for CD28 costimulation during TCR engagement and the outcome is different between CD4$^+$ and CD8$^+$ T cells.

The ectonucleotidase CD73 is recently recognized as an "immune checkpoint" and mediates adenosine generation[29]. Extracellular adenosine has been considered one of the main immunosuppressive factors in the tumor microenvironment. Its presence inhibits anti-tumor T cell responses[30]. Therefore, uncovering the source of adenosine production is crucial for the establishment of potentially effective therapies against cancer. CD73 is detected in tumor cells isolated from cancer patients[31,32], as well as in tumor cell lines[33]. Thus it is widely believed that tumor cells are a source of adenosine in the tumor microenvironment. It is noted that a higher percentage of CD28$^-$CD8$^+$ T cells in the total tumor-infiltrating CD8$^+$ T cell population is associated with advanced cancer staging and poor survival[13,14]. However, it has never been clarified how recruitment of CD28$^-$CD8$^+$ T cells advances cancer staging and whether CD28$^-$CD8$^+$ T cells contribute to the generation of extracellular adenosine in the tumor microenvironment. In this study, we found that mice inoculated with CD73-negative B16.gp33 tumor cells (Supplementary Fig. 3) have significantly higher CD73 on CD28$^-$CD8$^+$ T cells than on CD28$^+$CD8$^+$ T cells that were isolated from the draining lymph nodes (Fig. 5a), as well as from tumor (Fig. 5b). Our study of the B16.gp33 tumor model points to the fact that both CD28$^-$CD8$^+$ and CD28$^+$CD8$^+$ T cells exist in draining lymph nodes and solid tumors and that CD73-expressing CD28$^-$CD8$^+$ T cells can be a source of adenosine. In patients with colon cancer, we found that

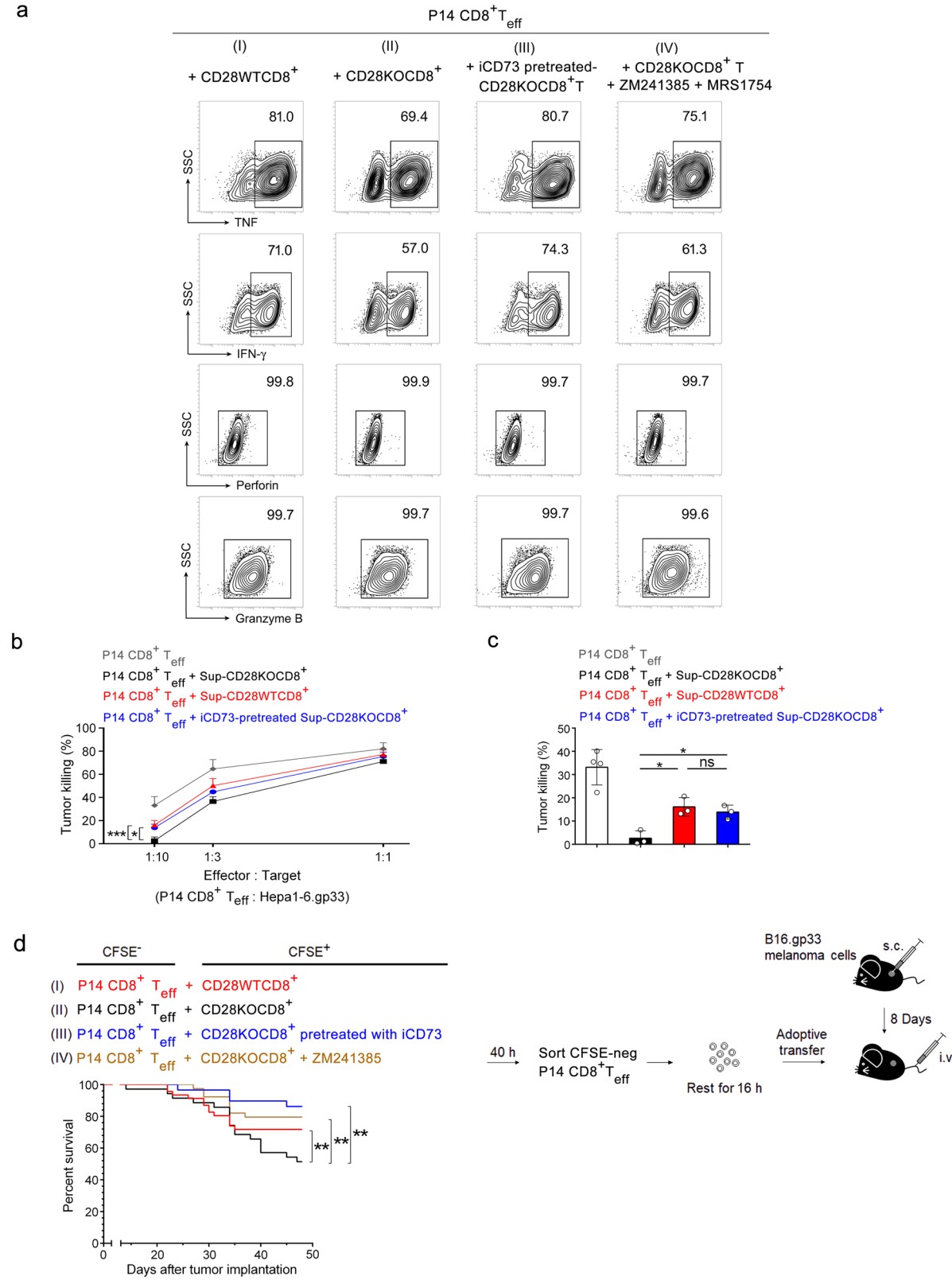

tumor-infiltrating CD8$^+$ T cells, as well as CD8+ T cells in peripheral blood express CD73 but CD73 expression on CD28$^-$CD8$^+$ T cells, is significantly increased after cell infiltration into the tumor as compared to that on CD28$^+$CD8$^+$ T cells (Fig. 5c, d). It is our speculation that when stimulated by tumor antigen in the absence of CD28 signaling in the tumor microenvironment and in lymphoid tissues, CD8$^+$ T cells upregulate the expression of CD73.

These observations suggest that CD73 expression on CD28$^-$CD8$^+$ T cells in the tumor-bearing host renders CD8$^+$ T cells immunosuppressive. Tumor-infiltrating CD28$^-$CD8$^+$ T cells together with CD73-expressing tumor cells generate adenosine to suppress T$_{eff}$ cells, allowing tumors to escape immune attack.

It is reported that interrupting B7-CD28 interaction blocks allograft rejection and ameliorate autoimmune responses

**Fig. 6 CD28 costimulation prevents CD73-mediated adenosine production by activated CD8$^+$ T cells. a** P14 CD8$^+$ T$_{eff}$ cells were pre-cultured with CFSE-labeled activated (I) CD28WTCD8$^+$; (II) CD28KOCD8$^+$; (III) iCD73-pretreated CD28KOCD8$^+$; (IV) CD28KOCD8$^+$ T cells in the presence of A$_{2A}$R antagonist ZM241385 and A$_{2B}$R antagonist MRS1754, at a ratio of 3:1. At 40 h after co-culture, CFSE-negative P14 CD8$^+$ T$_{eff}$ in each group was sorted, restimulated with PMA/ionomycin, and subjected to intracellular staining of TNF, IFN-γ, perforin, and granzyme B. Number on the upper right indicates the percentage of the protein-positive population. The data presented are representative of one of the three independent experiments. **b** P14 CD8$^+$ T cells treated with supernatants from activated CD28KOCD8$^+$ (black), CD28WTCD8$^+$ (red) or iCD73-pretreated CD28KOCD8$^+$ (blue) cell cultures were added to Hepa 1.6.gp33 target at E:T ratios of 1:10, 1:3, and 1:1 ($n = 3$ each). P14 CD8$^+$ T$_{eff}$ cells without any treatment were used as control (E:T ratios of 1:10, $n = 4$; 1:3, $n = 6$; 1:1, $n = 4$) (gray). **c** Further analysis of tumor-killing at E:T ratio of 1:10 in **b**. (P14 CD8$^+$ T$_{eff}$ cells without any treatment, $n = 4$; the other groups, $n = 3$). **d** Survival of tumor-bearing mice. On day 8 after tumor implantation, tumor-bearing mice were intravenously administered P14 CD8$^+$ T$_{eff}$ cells that had been pre-cultured with: (I) activated CD28WTCD8$^+$ (71.7% survival, $n = 46$, red); (II) activated CD28KOCD8$^+$ (51.4% survival, $n = 35$, black), significantly reduced survival between days 30 and 48 compared to mice in group I, $p < 0.01$; (III) iCD73-pretreated activated CD28KOCD8$^+$ (86.2% survival, $n = 29$, blue), (IV) activated CD28KOCD8$^+$ T cells in the presence of ZM241385 (79.5% survival, $n = 39$, brown). Statistical evaluations were performed using the Student's $t$-test with data expressed as the mean ± standard error of the mean (**c**). The ability of CD8$^+$ T cells between two groups to in vitro kill tumor at different E:T ratio was analyzed by the general linear model (**b**). The survival difference between groups was analyzed by log-rank test (**d**). *$p < 0.05$, **$p < 0.01$, ***$p < 0.001$; ns, not statistically significant.

through downregulation of T cell activation[34]. The results of our study indicate that upon TCR stimulation, the absence of CD28 signaling renders CD8$^+$ T cells immunosuppressive. Our data provide another mechanistic consideration for transplantation tolerance that suppressive CD8$^+$ T cells generated through insufficient CD28 costimulatory signaling in the graft tissue microenvironment may be the cause for graft tolerance. The relationship between CD28 signaling and CD73-mediated adenosine production we uncovered in this study makes CD28 costimulation a tempting drug target to regulate T cell function. This strategy may very well be applicable not only for cancer therapy but also transplantation tolerance and treatment of autoimmune disease.

In summary, our study defines a role for CD28 costimulation in CD8$^+$ T cell activation. CD28 engagement regulates CD73 expression and prevents CD8$^+$ T cells from becoming suppressive. Since CD28$^-$CD8$^+$ T cells are present in the tumor microenvironment, it is conceivable that CD73-mediated adenosine production by CD28$^-$CD8$^+$ T cells suppresses CD8$^+$ effector T cells to allow immune escape. This study points to the possibility of manipulating CD28$^-$CD8$^+$ T cells to develop anti-cancer therapeutics. Given the considerable interest in inducing suppressive T cells as a therapeutic strategy for autoimmune diseases and organ transplantation, our finding also unveils the CD28 costimulatory pathway as a potential target.

## Methods

**Animals**. Male C57BL/6 and P14 TCR transgenic mice[35,36] (P14, originally obtained from Dr. J. Kung, Academia Sinica) at age of 6–8 weeks were obtained from the animal center at National Taiwan University Hospital. CD28KO mice were acquired from the Jackson Laboratory. P14CD28KO mice were generated by crossing P14 mice to CD28KO mice. Animals were bred and housed in specific-pathogen-free conditions at the animal center at National Taiwan University Hospital, according to international guidelines on the care and use of laboratory animals. CD73KO mice were acquired from the Jackson Laboratory and bred and housed in specific-pathogen-free conditions at National Laboratory Animal Center, National Applied Research Laboratories, Taiwan. All animal experiments were performed following the guideline of the Use of Laboratory Animals published by NTU and approved by the Institutional Animal Care and Use Committee of College of Medicine and College of Public Health of NTU and were performed in compliance with the 3R principle (IACUC #20100131).

**Human samples**. Blood and primary tumor samples were obtained from patients with non-pretreated colon cancer at National Taiwan University Hospital. All of these samples were obtained after approval from the Institutional research ethics committee of National Taiwan University Hospital (protocol number: 202009044RINA, 12th October 2020) before its commencement and patients' written informed consent.

**Antibodies and reagents**. Anti-mouse CD3 (clone 145-2C11) and anti-CD28 (clone PV-1) antibodies were prepared in our laboratory. DMEM, penicillin and streptomycin from GIBCO Inc. (Grand Island, NY, USA); fetal bovine serum (FBS)

from HyClone Inc. (Logan, UT, USA); anti-mouse antibodies including Alexa 488 anti-Foxp3, -IFNγ and -perforin, PE anti-CD25, -CD39, -CD101, -CTLA-4, -granzyme B, ICOS and -IL-2, PerCP-Cy5.5 anti-CD73 and APC anti-CD45.2, FITC anti-human CD8, PE anti-human CD28, APC anti-human CD39 and PE-Cy7 anti-human CD73 from eBioscience (San Diego, CA, USA); PE anti-TNFα, -CCR6, -CD103, -Galectin-9 and -Helios antibodies from BioLegend (San Diego, CA, USA); PE anti-CD122, -CTLA-4, -LAG3 and -PD-1, APC anti-FR4, and PE-Cy7 anti-GITR antibodies from BD Bioscience (San Jose, CA, USA); Anti-CD73 antibody from Abcam (Cambridge, MA, USA); Mitomycin C, LPS, MTT, α, β-methylene adenosine 5'-diphosphate (AMPCP) from Sigma (St. Louis, MO, USA); KC7F2 from Calbiochem (Burlington, MA, USA); ZM241385 and MRS1754 from Tocris (Minneapolis, MN, USA); CFSE from Molecular Probes (Eugene, OR, USA); M2 peptides (LCMV gp33-41, KAVYNFATM) from AnaSpec, Inc. (San Jose, CA, USA); LY294002 from Cell signaling Technology (Danvers, MA, USA) were purchased.

**Naïve CD8$^+$ enrichment, cell sorting, and flow cytometry**. CD8$^+$ T cells from spleens of mice were enriched by positive selection using a magnetic bead kit (Miltenyi Biotec Inc., Bergisch Gladbach, Germany) prior to activation. CD62L$^{hi}$CD44$^{lo}$CD8$^+$ were sorted (FACSAria (BD Bioscience, San Jose, CA, USA) and used as naïve CD8$^+$ T cells (purity ≥ 99%) throughout the experiment. through service provided by Flow Cytometric Analyzing and Sorting Core Facility (First Core Laboratory, NTU, College of Medicine).

To determine the expression of surface antigens, live cells were stained by indicated fluorochrome-conjugated antibodies for 20–30 min on ice. Detection of intracellular cytokines was performed after 6-h stimulation with phorbol myristate acetate (PMA, 10 ng/mL)/Ionomycin (1 μg/mL) and 4-h culture in Brefeldin A (10 μg/mL). Live cells were fixed and permeabilized with cytofix-cytoperm kit (BD Biosciences) for staining of intracellular proteins. Intracellular Foxp3 was detected by Alexa 488 anti-Foxp3 (clone FJK-16s, eBioscience) after fixation and permeabilization using a Foxp3 staining buffer kit (eBioscience). BD FACSVerse™ (BD Biosciences) was employed for cell acquisition and BD FACSuite™ software for data analysis.

**CD8$^+$ T cell stimulation and culture conditions**. Cells culture was set up in DMEM containing 10% FBS and $5 \times 10^{-5}$ M 2-Mercaptoethanol. To activate CD8$^+$ T cells, naïve T cells (CD62L$^{hi}$CD44$^{lo}$CD8$^+$) were stimulated with anti-CD3 or anti-CD3/28 antibodies for 72 h and harvested. CD8$^+$ T$_{eff}$ used in this study were from naïve CD28WTCD8$^+$ T cells stimulated by plate-bound anti-CD3/28 antibodies. To generate P14 CD8$^+$ T$_{eff}$ cells, activate P14 CD8$^+$ T cells ($1 \times 10^5$ cell/mL) were activated by M2 antigenic peptide (LCMV gp33-41, KAVYNFATM, 0.08 nM, AnaSpec, Inc., San Jose, CA) presented by LPS-stimulated B6 B blast cells[37] (mitomycin C-treated[38], $5 \times 10^5$ cell/mL). At the end of the 3-day activation culture, the activated cells were washed and cultured in the presence of rhIL-2 (5 ng/mL) for 3–4 more days under 5% CO$_2$.

**Cytokine measurements**. Supernatants from activated CD8$^+$ T cell cultures were centrifuged at $400 \times g$ at 4 °C for 5 min, followed by $15,000 \times g$ at 4 °C for 10 min, and subjected to cytometric bead array for cytokine measurement. Cytokines were simultaneously quantified from 30 μL of culture supernatants using MILLIPLEX MAP Mouse Cytokine/Chemokine Magnetic Bead Panel. Data acquisition was performed according to the manufacturer's instruction using the Luminex 100 TM (Luminex) with xPONENT® software. MILLIPLEX® Analyst 5.1 Software was used to analyze and calculate cytokine concentrations in the supernatants.

**Collection of activated CD8$^+$ T cells-derived culture supernatants**. CD8$^+$ T cell suspensions were centrifuged at $400 \times g$ at 4 °C for 5 min, followed by $15,000 \times g$ at 4 °C for 10 min. The supernatants were collected for further experiments.

**Transwell experiments**. Activated CD28WTCD8$^+$ or CD28KOCD8$^+$ T cells and P14 CD8$^+$ T$_{eff}$ cells were added to the insert of the transwell culture device (Corning Inc., Corning NY, USA) and the bottom well separately at a cell ratio of 1:3. P14 CD8$^+$ T$_{eff}$ cells were harvested 40 h later, and subject to tumor cell lysis assay or for adoptive transfer to B16.gp33 melanoma tumor-bearing mice.

**T cell-mediated tumor cell lysis assay**. P14 CD28WTCD8$^+$ T$_{eff}$ cells were added to Hepa 1-6.gp33 cell culture at the indicated E:T ratio (P14 CD28WTCD8$^+$ T$_{eff}$: Hepa 1-6.gp33) of 1:100, 1:30 and 1:10 for experiments of addition of culture supernatants, and 1:3, 1:1 and 3:1 for transwell experiments. At 14 h after co-culture, the viability of Hepa 1.6.gp33 cells was assessed by MTT (3-(4,5-Dime-thylthiazol-2-yl)−2,5-diphenyltetrazolium bromide) assay. The percentage of target cells that were killed by P14 CD8$^+$ T$_{eff}$ was calculated by the formula: (total cells-viable cells)/total cells × 100%.

**B16.gp33 melanoma model with adoptive transfer of P14 CD8$^+$ effector cells**. B16.gp33 cells were derived from B16 melanoma cells and genetically modified to express gene encoding amino acid 33–41 of glycoprotein from lymphocytic choriomeningitis virus (LCMV) (kindly provided by Dr. Hanspeter Pircher)[39]. Cells were cultured in DMEM supplemented with 10% FBS and 200 µg/mL G418. Following subcutaneous inoculation of B16.gp33 cells ($1 \times 10^6$ cells/mouse), the diameter of the tumor and survival were recorded on the indicated day. P14 CD8$^+$ T$_{eff}$ cells specific for LCMV gp33 in the context of H-2D$^b$ were generated[40]. P14 CD8$^+$ T$_{eff}$ cells in 1× PBS ($1 \times 10^7$ cells/0.15 mL/mouse) were injected intravenously into mice that had been inoculated with B16.gp33 tumor 8 days previously.

**Isolation of cells from lymph nodes and tumors**. The inguinal lymph nodes were dissected from control and B16.gp33 tumor-bearing mice and placed in a sterile Petri dish containing 2 mL ice-cold 5% FBS/PBS. Lymph nodes were disrupted using two 25 G needles fixed on a 1 mL syringe. To isolate cells from tumors, B16. gp33 tumors were dissociated using GentleMACS mechanical system according to the manufacturer's protocol. Cell suspensions from lymph nodes, as well as from tumors were filtered through a 70 µm pore-size screen (BD Biosciences) and washed with 5% FBS/PBS before staining and flow cytometric analysis.

**Isolation of peripheral blood mononuclear cell (PBMC) and tumor-infiltrating lymphocytes (TILs) from patients with colon cancer**. PBMCs were purified from the blood of patients with non-pretreated colon cancer by Ficoll density gradient. TILs of colon cancer were obtained by dissociating tumors using GentleMACS mechanical system according to the manufacturer's protocol.

**Treatment of inhibitors and antagonists**. To inhibit PI3K, a specific inhibitor, LY294002 (3 µM), was added upon stimulation of naïve CD28WTCD8$^+$ T cells by anti-CD3/28. To block CD73 activity of activated CD28KOCD8$^+$ T cells, cells were pretreated with a selective and competitive inhibitor of CD73, AMPCP (100 µM) for 24 h and harvested for further experiments. To block the effect of adenosine on CD8$^+$ T$_{eff}$ cells, A$_{2A}$ receptor antagonist, ZM241385 (0.5 µM) and/or A$_{2B}$ receptor antagonist, MRS1754 (0.5 µM), were added to the CD28KOCD8$^+$ T cell/CD8$^+$ T$_{eff}$ cell co-culture at a final concentration of 0.5 µM.

**Quantitation of adenosine through ultra-performance liquid chromatography system coupled online to the Waters Xevo TQ-S triple quadrupole mass spectrometer (UPLC-MS/MS)**. The capacity of CD8$^+$ T cells to degrade AMP to adenosine was analyzed after 2-h incubation at 4 °C for under 5% CO2 with labeled AMP$_{13C,15N}$ (from Sigma-Aldrich) in 200 mL of serum-free IL-2-containing DMEM medium supplemented with antibiotics and L-glutamine (Life Technologies)[41]. In some cases, cells were preincubated with AMPCP (100 µM) for 30 min before the experiment. Cell supernatants were harvested for further quantification of adenosine and AMP. Levels of endogenous generation of extracellular adenosine and AMP were measured by quantification of culture supernatants without the addition of AMP$_{13C,15N}$. The LC system used for analysis was ultra-performance liquid chromatography (UPLC) system (ACQUITY UPLC, Waters, Millford, MA). The sample was separated with ZIC-pHILIC column (5 µm particle size, $2.1 \times 100$ mm, Merck-Millipore). The UPLC system was coupled online to the Waters Xevo TQ-S triple quadrupole mass spectrometer. The flow rate was 0.2 mL/min, the injection volume of 2 µL, the column temperature of 25 °C. The composition of mobile phase A was water containing 20 mM ammonium carbonate, B was Acetonitrile, and gradient: 0 min-20%A, 2 min-30%A, 5 min-40%A, 6 min-60%A, 7 min-60%A, 8 min-95%A, 10 min-95%A, 11 min-20% A, 15 min-20%A. Characteristic MS transitions were monitored using positive multiple reaction monitoring (MRM) mode for AMP ($m/z$, 348 > 136), $^{13}C_{10},^{15}N_5$AMP ($m/z$, 363 > 146), adenosine ($m/z$, 268 > 136), and $^{13}C_{10},^{15}N_5$ adenosine ($m/z$, 283 > 146). Data acquisition and processing were performed using MassLynx version 4.1 and TargetLynx software (Waters Corp.).

**Statistics and reproducibility**. Experiments were performed independently at least three times. The amount of cytokine and adenosine production in culture supernatants, relative MFI of cytokine production in CD8$^+$ T$_{eff}$ cells, relative MFI of CD73 protein, and fold of tumor size, between groups, were analyzed by Student's $t$-test using GraphPad Prism version 6 software (GraphPad Software, La Jolla, CA, USA). The ability of CD8$^+$ T cells between two groups to in vitro kill tumor at different E:T ratio was analyzed by a general linear model using the SAS software version 9.4 (SAS Institute, Cary, NC). The survival difference between groups was analyzed by log-rank test using the SAS software version 9.4. (SAS Institute, Cary, NC). Statistical significance was set: *$p < 0.05$, **$p < 0.01$, ***$p < 0.001$.

**Reporting summary**. Further information on research design is available in the Nature Research Reporting Summary linked to this article.

## Data availability
All data needed to evaluate the conclusions in the paper are present in the paper or the Supplementary Materials. Source data underlying plots shown in figures are provided in Supplementary Data 1. The data used to generate and support this study will be available from the corresponding author upon request.

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

## Acknowledgements

We are grateful to Yu-Chia Su for animal experiment technical support. We thank Ching-Pin Chang, I-Hsin Su, and Chih-Yung Tang for helpful discussions. We also thank Pei-Ru Wang and Shih-Fu Wu for their technical assistance. We appreciate the service provided by the Flow Cytometric Analyzing and Sorting Core Facility of the First Core Laboratory, College of Medicine, National Taiwan University. We thank Ting-Hsiang Chang for UPLC-MS/MS parameter optimization and Metabolomics Core Facility, Agricultural Biotechnology Research Center at Academia Sinica, for technical support. The work was supported by grants from the Ministry of Science and Technology (NSC 99-2314-B-002-081-MY3), National Taiwan University Hospital (MS-353), and Liver Disease Prevention and Treatment Research Foundation, Taiwan.

## Author contributions

Y.P.L. designed and performed most of the experiments and analyzed the data. Y.P.L., L.C.K., H.T.C., P.C.I.K., and H.N.H. contributed to experimental design and interpretation of results. Y.P.L., B.R.L., H.J.L., C.Y.L., Y.T.C., P.W.H., H.C.C., H.Y.C., J.L., B.A.W.-H., and J.T.K. conducted experimental procedures and analyzed data. S.C.C. had the overall responsibility of project conceptualization, experimental design, investigation, data analysis, and validation, performed some experiments, and wrote the manuscript.

## Competing interests

The authors declare no competing interests.
