## [Peer Review File · Communications Biology]

Reviewer #1 (Remarks to the Author):

This very interesting article by Chen et al. shows the differential ability of CD28+ vs. CD28- CD8+ T cells to suppress the ability of P14 effector T cells to control tumor growth using B16.gp33 tumor-bearing mice. The experimental design is somewhat complicated. CD28+ vs. CD28- CD8+ T cells are activated with anti-CD3/anti-CD28 and then cultured with activated P14 effector T cells for 36 hours in a ratio of 1:3. The effector T cells are then used in in vitro killing assays or adoptively transferred into B16.gp33 tumor-bearing mice. The authors conclude that the reduced killing activity of P14 effector T cells (and their reduced cytokine production) precultured with CD28- CD8+ T cells is caused by the upregulation of CD73 expression that occurs on CD28- CD8+ T cells during priming. Flow cytometry confirms that CD73 expression is higher on CD28- compared to CD28+ CD8+ T cells after activation with anti-CD3/anti-CD28. The authors posit that CD73 catalyzes the production of extracellular adenosine that then inhibits the cytokine production and tumor-killing ability of adoptively transferred P14 Teff cells. This conclusion is consistent with their observation that supernatants from activated CD28- CD8+ T cells are also immunosuppressive. The authors propose that CD28 costimulation inhibits the upregulation of CD73 on CD28+ CD8+ T cells, resulting in less immunosuppression. This conclusion is supported by the observation that CD73 upregulation can be restored with an inhibitor of CD28 signaling.

Major criticisms:

The data are largely consistent with the hypotheses presented by the authors. However, there are some pieces of data that are unexpected and suggest that the story may not be so simple. First, the authors measured adenosine levels in the supernatants of CD28+ vs. CD28- CD8+ T cells and found them to be 100 μM vs. 240 μM . These levels are unexpectedly high by 10- to 100-fold. Physiological levels of adenosine are in the low μM range. While it is true that CD73 can produce extracellular adenosine, it does so only in the presence of extracellular AMP. What is the source of AMP in the authors' experimental system? It is very easy to get falsely elevated levels of adenosine if there is any cell death in the experimental system, as the concentration of intracellular ATP is in the mM range. Any ATP released by dying cells can be rapidly converted to AMP, and if CD73 is present, also to adenosine. Even 1% dead cells can lead to falsely elevated adenosine levels. Furthermore, adenosine is rapidly degraded by adenosine deaminase that is present in fetal bovine serum. It is normally difficult to achieve elevated levels of adenosine in cell culture of CD73+ cells without either adding exogenous AMP and/or an inhibitor of adenosine deaminase. Therefore, one has to be suspicious of the adenosine levels reported in this manuscript. The best way to prove that CD73-generated adenosine is being produced is to compare results with T cells from wild type and CD73 KO mice that can be obtained from Jackson Laboratories. Even this control will not eliminate the possibility that the high levels of adenosine claimed in this paper are generated from ATP released from dying cells. Another problem is that the A2a adenosine receptor has a high affinity for adenosine and can be activated by concentrations in the low μM range. Therefore, the concentration of adenosine allegedly produced by CD28+ CD8+ T cells should be more than enough to engage the A2a receptor and suppress the function of Teff.

Before claiming that the results of their studies can be used to predict therapies for human cancer, the authors should discuss what is known about CD73 expression on human CD8+ cells. In humans, about 50% of CD8+ T cells express high levels of CD73, while the other 50% are CD73-. Therefore, production of adenosine in the tumor microenvironment by CD73+ CD8+ T cells could potentially play a role in creating an immunosuppressive milieu. This idea is not new. However, whether this is regulated by CD28 signaling is another question altogether.

Minor criticism:

Another aspect of the data that is troubling is the results showing downregulation of CD73 expression with the HIF-1 α inhibitor, KC7F2 in Figure 6a. While it is true that the hypoxia-induced upregulation of CD73 is mediated by a HIF-1 α binding site in the CD73 promoter, there is no evidence that CD73 expression is regulated by HIF-1 α under normoxic conditions (see ref. 25, cited by the authors).

It is unclear why the authors used such a complicated method for adenosine measurements. If the concentrations of adenosine are truly in the 100 μ M range, they would be easily detected by HPLC, a method that could be more easily reproduced by other investigators.

Reviewer #2 (Remarks to the Author):

This study delineates the role of CD28 co-stimulation in immunoregulation of T cells via CD73 expression. Importantly, authors show that CD28 engagement during T cell activation differs in CD4+ and CD8+ T cells. This study sheds light on the mechanism of how CD28- T cells found in tumor microenvironment aid in immune escape of tumor cells. This is an interesting study and provides a perspective for the potential use of CD28 pathway as a therapeutic target. However, there are a few points that the authors need to address for consideration of publication of this manuscript.

1. In the experiment shown in Fig. 1e and 1i, it is very obvious that the error bars of the Supernatant-CD28WTCD8+ data set shown in red, are overlapping with the other data set. This raises a concern as to how the authors performed statistical analysis on this experiment and how it could lead to a highly significant difference despite the huge error on every E:T ratio. It is also not clear as to why the authors chose to only show the error bars in a single direction. Authors need to provide the raw data to the reviewers and explain how the statistical analysis was done. These pieces of data weaken the otherwise extensive work the authors have presented in the remaining manuscript.
2. Although briefly mentioned in the abstract, authors need to clearly describe the novelty of their study and exactly what gap it fills in the field, maybe in the discussion section. This is lost in the intricate details of the manuscript.
3. Details about the experimental systems, for example, the P14 transgenic mice, M2 peptide stimulation etc. need to be introduced before the results section, considering the wide readership this paper may have.

Re: Manuscript COMMSBIO-20-1079-T by Lai et al. entitled “CD28 engagement inhibits CD73-mediated regulatory activity of CD8⁺ T cells”

Responses To Reviewer 1:

Major criticisms:

1. First, the authors measured adenosine levels in the supernatants of CD28⁺ vs. CD28⁻ CD8⁺ T cells and found them to be 100 μ M vs. 240 μ M. These levels are unexpectedly high by 10- to 100-fold. Physiological levels of adenosine are in the low μ M range. While it is true that CD73 can produce extracellular adenosine, it does so only in the presence of extracellular AMP. What is the source of AMP in the authors 19; experimental system? It is very easy to get falsely elevated levels of adenosine if there is any cell death in the experimental system, as the concentration of intracellular ATP is in the mM range. Any ATP released by dying cells can be rapidly converted to AMP, and if CD73 is present, also to adenosine. Even 1% dead cells can lead to falsely elevated adenosine levels.

Response: We re-did all relevant experiments and employed highly reliable new methodology in detecting adenosine levels in T cell culture supernatants. (A) The levels of adenosine were measured by ultra-performance liquid chromatography system coupled online to the Waters Xevo TQ-S triple quadrupole mass spectrometer (UPLC-MS/MS). (B) Supernatants were collected after 2 h incubation instead of 24 h as in the original manuscript. (C) Cells were cultured in serum-free DMEM medium containing IL-2. Results are shown in **Fig. 4g** and **4h**. The levels of adenosine are in μ M range.

Fig. 4g:

Fig. 4h:

We also addressed the concern of falsely elevated levels of adenosine attributed by dead cells: (A) We added saturated exogenous AMP_{13C,15N} isotope (37.5 μ M) to anti-

CD3- and anti-CD3/28-antibody-stimulated CD28WTCD8⁺ T cell cultures and measured adenosine^{13C,15N} levels by UPLC-MS/MS. The levels of CD73-mediated production of adenosine were significantly higher in the supernatants of anti-CD3-antibody-stimulated CD28WTCD8⁺ T cells ($13.70 \pm 1.149 \mu\text{M}$) than anti-CD3/28-antibodies stimulated CD28WTCD8⁺ T cells ($6.27 \pm 0.716 \mu\text{M}$) (**Fig. 4h**, $p = 0.001$), demonstrating that CD73 activity is higher in anti-CD3 antibody-stimulated cells than cells with anti-CD3/28 antibody treatment. (B) We also used CD73-knockout CD8⁺ T cells as a control. In the absence of CD73, there was no difference in the levels of residual AMP between anti-CD3 and anti-CD3/28 antibody-stimulated cells regardless the source of AMP was endogenous (**supplementary Fig 2c**) or exogenous (**supplementary Fig 2e**). Results of this experiment demonstrate that conversion of AMP to adenosine is mediated by CD73. Even there was dead cells in the culture, their numbers were probably comparable between cells treated with anti-CD3 and anti-CD3/28 antibodies. (C) We collected culture supernatants after 2 h incubation, a time when cell death may be marginal and AMP to adenosine conversion is taking place. These conditions were adopted from a published paper ¹.

Supplementary Fig 2c:

Supplementary Fig 2e:

2. Adenosine is rapidly degraded by adenosine deaminase that is present in fetal bovine serum. It is normally difficult to achieve elevated levels of adenosine in cell culture of CD73⁺ cells without either adding exogenous AMP and/or an inhibitor of adenosine deaminase. Therefore, one has to be suspicious of the adenosine levels reported in this manuscript. The best way to prove that CD73-generated adenosine is being produced is to compare results with T cells from wild type and CD73 KO mice that can be obtained from Jackson Laboratories. Even this control will not eliminate the possibility that the high levels of adenosine claimed in this paper are generated from ATP released from dying cells.

Response: To avoid the action of adenosine deaminase that is present in fetal bovine serum, we cultured cells in serum-free medium containing IL-2 and collected supernatants at 2 h after incubation. The levels of adenosine in cell culture supernatants were quantified by UPLC-MS/MS. Following the reviewer's suggestion, we also obtained CD73 KO mice from Jackson Laboratories and showed that the adenosine we detected in the supernatants is generated through the activity of CD73. Results are shown in **Fig. 4g** and **4h**.

3. Another problem is that the A2a adenosine receptor has a high affinity for adenosine and can be activated by concentrations in the low μM range. Therefore, the concentration of adenosine allegedly produced by CD28⁺ CD8⁺ T cells should be more than enough to engage the A2a receptor and suppress the function of T_{eff}.

Response: We agree with Reviewer's point that "the A2a adenosine receptor has a high affinity for adenosine". Our study did not exclude the possibility that CD28⁺CD8⁺ T cells can produce adenosine. However, upon activation, CD73 expression is significantly higher in CD28⁻CD8⁺ T cells than in CD28⁺CD8⁺ T cells. The level of adenosine is also higher in CD8⁺ T cells stimulated with anti-CD3 antibody than cells stimulated with anti-CD3/28 antibodies. A logical conclusion is that stimulation of CD8⁺ T cells through TCR in the absence of CD28 signaling, CD28⁻CD8⁺ T cells effected higher suppressive activity than CD28⁺CD8⁺ T cells.

4. The authors should discuss what is known about CD73 expression on human CD8⁺ cells. In humans, about 50% of CD8⁺ T cells express high levels of CD73, while the other 50% are CD73⁻. Therefore, production of adenosine in the tumor microenvironment by CD73⁺ CD8⁺ T cells could potentially play a role in creating an immunosuppressive milieu.

Response: We followed up on Reviewer's suggestion to study CD73 in CD8⁺ T cells from cancer patients. PBMC and tumor-infiltrating lymphocytes (TILs) were collected from patients with colon cancer and the CD73 expression on CD28⁻CD8⁺ T cells and CD28⁺CD8⁺ T cells in both circulation and tumor were analyzed. We found that CD73 expression in CD28⁻CD8⁺ T cells isolated from tumor had significantly higher levels of CD73 than CD28⁻CD8⁺ T cells in the circulation (MFI TIL/PBMC ratio = 4.44 ± 0.822) ($p < 0.0001$) while CD73 expression in tumor-infiltrating CD28⁺CD8⁺ T cells was

slightly upregulated compared to CD28⁺CD8⁺ T cells in the peripheral blood (MFI TIL/PBMC ratio = 1.61 ± 0.226) (**Fig. 5c**). We further did paired analysis of CD73 expression on CD28⁺CD8⁺ and CD28⁻CD8⁺ T cells in peripheral blood and in tumor from the same individual. Results in **Figure 5d** show that there was significantly higher CD73 upregulation on CD28⁻CD8⁺ T cells than that on CD28⁺CD8⁺ when they were both in the tumor microenvironment. We added a paragraph to Discussion (line 329-356) to discuss how CD73 expression on human CD28⁻CD8⁺ cells contributes to adenosine production in the tumor microenvironment.

Fig. 5c:

Fig. 5d:

Minor criticisms:

1. Another aspect of the data that is troubling is the results showing downregulation of CD73 expression with the HIF-1 α inhibitor, KC7F2 in Figure 6a. While it is true that the hypoxia-induced upregulation of CD73 is mediated by a HIF-1 α binding site in the CD73 promoter, there is no evidence that CD73 expression is regulated by HIF-1 α under normoxic conditions (see ref. 25, cited by the authors).

Response: We agree with the reviewer. We removed figure 6a (in the original manuscript) that described HIF-1 α -induced upregulation of CD73 in CD8⁺ T cells in the absence of CD28 costimulation in the revised manuscript.

Reference:

1 Gourdin, N. *et al.* Autocrine Adenosine Regulates Tumor Polyfunctional CD73(+)/CD4(+) Effector T Cells Devoid of Immune Checkpoints. *Cancer Res* **78**, 3604-3618, doi:10.1158/0008-5472.CAN-17-2405 (2018).

Re: Manuscript COMMSBIO-20-1079-T by Lai et al. entitled “CD28 engagement inhibits CD73-mediated regulatory activity of CD8⁺ T cells”

Responses To Reviewer 2:

Re: Manuscript COMMSBIO-20-1079-T by Lai et al. entitled “CD28 engagement inhibits CD73-mediated regulatory activity of CD8⁺ T cells”

Point 1. (A) In the experiment shown in Fig. 1e and 1i, it is very obvious that the error bars of the Supernatant-CD28WTCD8⁺ data set shown in red, are overlapping with the other data set. This raises a concern as to how the authors performed statistical analysis on this experiment and how it could lead to a highly significant difference despite the huge error on every E:T ratio. (B) It is also not clear as to why the authors chose to only show the error bars in a single direction. (C) Authors need to provide the raw data to the reviewers and explain how the statistical analysis was done. These pieces of data weaken the otherwise extensive work the authors have presented in the remaining manuscript.

Response: (A) We used general linear models (GLM) analysis to compare the overall difference in the ability of CD8⁺ T cells to kill tumor cells between the two groups at different effector:target ratios in Figures 1e and 1i. General linear models, $E(Y) = X\beta, V(Y) = \sigma^2 I$ were constructed. “Y” indicated the matrix of percentage of killed tumor cells; “X”, the design matrix including two groups of CD8⁺ T cells at different effector:target ratios; “ β ”, the parameters; “I”, the identity matrix; “ σ^2 ”, the common variance for the errors. Type III sums of squares were used for hypothesis testing. The statistical analysis of general linear models was performed using the SAS software. (B) The error bar is now bidirectional. (C) Raw data for **Figure 1e** and **1i** are shown below.

Fig. 1e:

Fig. 1i:

Figure 1e: presented significant difference in tumor cell killing effect between the Sup-CD28WTCD8⁺- and Sup-CD28KOCD8⁺-treated P14 CD8⁺ T_{eff} cells ($p = 0.0016$)

The numbers in the table indicate percentage of tumor cell death.

P14 CD8⁺ T_{eff} + Sup-CD28WTCD8⁺:

	1 st experiment	2 nd experiment	3 rd experiment	4 th experiment
E:T ratio 1:100	7.3	12.4	20.9	28.9
E:T ratio 1:30	20.3	22.2	24.2	35.3
E:T ratio 1:10	24.0	34.2	36.8	-

P14 CD8⁺ T_{eff} + Sup-CD28KOCD8⁺:

	1 st experiment	2 nd experiment	3 rd experiment
E:T ratio 1:100	1.1	-3.9	3.5
E:T ratio 1:30	15.9	7.3	9.7
E:T ratio 1:10	23.1	27.3	16.0

Figure 1i: presented significant difference in tumor cell killing effect between the Sup-CD28WTCD8⁺- and Sup-CD28KOCD8⁺-exposed P14 CD8⁺ T_{eff} cells ($p = 0.0001$)

The numbers in the table indicate percentage of tumor cell death.

P14 CD8⁺ T_{eff} exposed to CD28WTCD8⁺ Sup:

	1 st experiment	2 nd experiment	3 rd experiment	4 th experiment	5 th experiment
E:T ratio 1:3	34.1	36.3	41.7	42.6	-
E:T ratio 1:1	41.8	58.0	81.6	86.0	-
E:T ratio 3:1	91.3	93.3	92.6	79.6	76.4

P14 CD8⁺ T_{eff} exposed to CD28KOCD8⁺ Sup:

	1 st experiment	2 nd experiment	3 rd experiment	4 th experiment	5 th experiment	6 th experiment
--	-------------------------------	-------------------------------	-------------------------------	-------------------------------	-------------------------------	-------------------------------

E:T ratio 1:3	20.9	34.3	26.1	21.5	24.9	26.2
E:T ratio 1:1	44.0	44.9	40.3	31.4	36.0	32.0
E:T ratio 3:1	85.7	88.2	79.7	48.9	51.9	48.6

Point 2. *Although briefly mentioned in the abstract, authors need to clearly describe the novelty of their study and exactly what gap it fills in the field, maybe in the discussion section. This is lost in the intricate details of the manuscript.*

Response: We rewrote the abstract and added the description of the novelty of this study in **Discussion** (line 329-356).

Point 3. *Details about the experimental systems, for example, the P14 transgenic mice, M2 peptide stimulation etc. need to be introduced before the results section, considering the wide readership this paper may have.*

Response: We added the details about P14 transgenic mice, M2 peptide stimulation, statistical analysis, UPLC-MS/MS method for adenosine analysis in Results and Methods in the revised manuscript (line 112-119, 438-443, 501-524, 529-531).

REVIEWERS' COMMENTS:

Reviewer #1 (Remarks to the Author):

The authors responded very carefully to my original critique and the manuscript is significantly improved. I found only a few minor errors. First in supplemental figure 1, there is no red line - it is blue. Second, some references were omitted on page 23, lines 330 and 332.

Reviewer #2 (Remarks to the Author):

The authors have addressed all the concerns raised during the initial review sufficiently.

Re: Manuscript COMMSBIO-20-1079-T by Lai et al. entitled “CD28 engagement inhibits CD73-mediated regulatory activity of CD8⁺ T cells”

Responses To Reviewer 1:

Reviewer #1's comment:

The authors responded very carefully to my original critique and the manuscript is significantly improved. I found only a few minor errors. First in supplemental figure 1, there is no red line - it is blue. Second, some references were omitted on page 23, lines 330 and 332.

Response:

We appreciate your critical review. We have corrected all the errors. (1) The figure legend of Supplementary figure 1 was corrected. (2) The two references that were omitted have been added properly.

Re: Manuscript COMMSBIO-20-1079-T by Lai et al. entitled “CD28 engagement inhibits CD73-mediated regulatory activity of CD8⁺ T cells”

Responses To Reviewer 2:

Reviewer #2's comment:

The authors have addressed all the concerns raised during the initial review sufficiently.

Response: We appreciate your critical review.